# Advanced machine learning approaches for predicting Neglected Tropical Disease co-endemicity in Kenya: A focus on soil-transmitted helminths, schistosomiasis, and lymphatic filariasis

Nkuba Nyerere[1], Damaris Felistus Mulwa[1,2]*

**1** Department of Mathematics and Statistics, Sokoine University of Agriculture, Morogoro, Tanzania,
**2** Department of Statistics and Actuarial Sciences, Jomo Kenyatta University of Agriculture and Technology, Nairobi, Kenya

* damaris.mulwa@jkuat.ac.ke

## Abstract

### Background

Neglected Tropical Diseases (NTDs) affect 1.5 billion people worldwide with 39% of the global burden occurring in Africa. In Kenya, NTDs remain endemic despite control efforts, with co-endemicity of soil-transmitted helminths (STH), schistosomiasis (SCH), and lymphatic filariasis (LF) complicating intervention strategies. This study developed machine learning models to predict high-risk co-endemic areas using demographic and Water, Sanitation, and Hygiene (WASH) indicators.

### Methodology

The study analyzed Kenya's 2022 NTD co-endemicity data from the Expanded Special Project for Elimination of Neglected Tropical Diseases, incorporating WASH and population variables. Three machine learning algorithms, Random Forest, Gradient Boosting Machine, and Extreme Gradient Boosting (XGBoost) were trained to classify regions by STH prevalence levels and co-endemicity status. Model performance was evaluated using cross-validation, Receiver Operating Characteristic – Area under the Curve (AUC) and variable importance analysis.

### Results

The RF model achieved the highest predictive performance (AUC = 0.70), followed by XGBoost (AUC = 0.66) and GBM (AUC = 0.62). Key predictors included improved sanitation access (mean importance score: 0.24), population density (0.21), and co-endemicity with LF/SCH (0.18). Spatial analysis identified Eastern and North-Eastern Kenya as persistent hotspots, correlating with low WASH coverage (<40%).

**Data availability statement:** The data used in this study is publicly available via the link https://espen.afro.who.int/maps-data/countries/kenya#overview--coendemicity.

**Funding:** The author(s) received no specific funding for this work.

**Competing interests:** The authors have declared that no competing interests exist.

## Conclusion

Machine learning models effectively identified the high-risk NTD co-endemic areas in Kenya, with RF outperforming other models. These findings support targeted interventions integrating WASH improvements with mass drug administration in identified hotspots. We propose a real-time dashboard for dynamic risk mapping to optimize resource allocation; a strategy aligned with Kenya's NTD Elimination Strategic Plan 2030.

### Author summary

NTDs are a diverse group of communicable diseases affecting 1.5 billion people globally. In Africa, where about 39% of the world's NTD burden occurs, this study effectively used machine learning (ML) models to identify, map and forecast high-risk areas for NTD co-endemicity. The study specifically examined the co-occurrence of lymphatic filariasis (LF), schistosomiasis (SCH), and soil-transmitted helminths (STH). The results show that machine learning models in particular, the random forest are useful instruments for dynamic risk mapping of regions where NTD co-endemicity is present. The findings highlight how important the Water, Sanitation, and Hygiene (WASH) indicators are in addition to conventional MDA whereas the spatial analysis identified Eastern and North-Eastern regions as the persistent hotspots, correlating with low WASH coverage (<40%). In accordance with Kenya's NTD Elimination Strategic Plan 2030, the study findings suggest putting in place a real-time dashboard for ongoing risk monitoring and surveillance in order to facilitate focused interventions and maximize resource allocation.

## 1. Introduction

Co-endemicity refers to the geographic overlap of two or more diseases in the same population [1]. NTDs are a diverse group of communicable diseases affecting 1.5 billion people globally [2]. Furthermore, the African population's bear 39% of the total global burden of NTDs and, of this, 90% is accounted for by five diseases (LF, Onchocerciasis, STH, SCH and Trachoma) which are treated through regular preventive chemotherapy (PC-NTDs) in a mechanism called mass drug administration [3]. Up to 44 countries in Africa are endemic for at least 1 PC-NTD, 42 for at least 2 PC-NTDs and 17 for all the 5 PC-NTDs [4]. These preventable and treatable diseases cause severe pain, stigma and disfigurement while killing over 170,000 people and costing developing economies billions of dollars every year [5]. In Africa, only eight countries have eliminated at least one PC-NTD [6]. Despite the intervention control strategies, Kenya remains endemic of LF, SCH and STH. These infections, especially the STH, remain endemic especially in the rural areas with inadequate water and sanitation supply [7]. Despite progress in mass drug administration campaigns, persistent transmission hotspots necessitate refined surveillance and intervention strategies.

LF is commonly known as elephantiasis and is a painful and profoundly disfiguring disease [8]. In endemic countries, the disease has a major social and economic impact [9]. It is primarily caused by three species of parasitic roundworms; *Wuchereria bancrofti* (the most common, and only species present in Africa), *Brugia malayi*, and *Brugia timori* [10]. These parasites are transmitted to humans through the bites of infected mosquitoes. Once inside the human body, the larvae develop into adult worms in the lymphatic system [11]. The key statistics for 2022 LF transmissions reveal that the population living in the implementation units was 4,257,611 people [3]. The population living in the implementation units and had stopped preventive chemotherapy was 319,806 while the population targeted for the preventive chemotherapy was 3,721,891.

STH are caused by parasitic worms and are widespread in areas with inadequate sanitation [12,13]. They primarily affect the intestines, leading to a range of health issues, with the most severe impacts on children [14]. Control and prevention efforts focus on improving hygiene and sanitation, as well as regular deworming treatments [15]. The main worms causing soil-transmitted helminthiases are roundworm (*Ascaris lumbricoides*), whipworm (*Trichuris trichiura*), and hookworms (*Necator americanus* and *Ancylostoma duodenale*) [16]. These parasites are prevalent in areas with poor sanitation and are primarily transmitted through soil contaminated with human feces. In Kenya, the key statistics for STH transmission in 2022 showed that the population living in the implementation units and requiring preventive chemotherapy was 1,884,837 pre-school children and 4,180,094 school age children. The targeted population for preventive chemotherapy was 590,377 pre-school children and 1,556,647 school age children [1]. The population requiring preventive chemotherapy had a geographic coverage of 27%.

SCH is also known as bilharzia and is caused by parasitic flatworms called schistosomes and is transmitted through contact with contaminated freshwater [17]. It can cause a wide range of chronic health problems, including damage to internal organs [17]. The main species causing schistosomiasis are *Schistosoma mansoni*, *Schistosoma haematobium*, and *Schistosoma japonicum* [18]. These species vary geographically and cause different forms of the disease. In Africa, the two most prevalent species are *S. mansoni* and *S. haematobium*. SCH is most prevalent in tropical and subtropical areas, especially in poor communities without access to safe drinking water and adequate sanitation [19]. It's estimated to affect millions of people worldwide. In 2022, the key statistics for SCH reveals that the population living in the implementation units and requiring preventive chemotherapy was 5,665,977 school age children and 3,351,000 adults. The total population targeted with preventive chemotherapy was 1,329,949 school age children and a geographic coverage of 32% required preventive chemotherapy [2].

The Kenya's Ministry of Health has implemented a Strategic Plan for control of NTDs through the National NTD control program [3]. STH transmission is strongly influenced by environmental and socioeconomic factors, including access to clean water, sanitation, and hygiene (WASH) practices [20]. It's co-endemicity with LF and SCH further exacerbates disease burden, as overlapping risk factors (e.g., poor sanitation, vector exposure) amplify transmission dynamics [21]. Current control efforts rely on geospatial mapping and periodic surveys, but these approaches may lack predictive precision for emerging high-risk zones. Machine learning offers a promising tool for enhancing NTD surveillance by identifying nonlinear relationships between risk factors and disease prevalence [21]. Recent studies have applied ML to predict schistosomiasis in Côte d'Ivoire [22] and LF in India [23], but few have integrated co-endemicity and WASH data for co-endemicity risk modeling in Kenya.

Despite the ongoing efforts to control and eliminate NTDs, current surveillance and intervention strategies often rely on static maps, outdated prevalence data, and non-predictive models. This limits the timely identification of high-risk areas and fails to capture the dynamic co-endemic interactions between the common NTDs like STH, SCH, and LF, which frequently co-occur in the same vulnerable populations, particularly in Sub-Saharan Africa. While supervised machine learning methods and models have been increasingly applied in broader disease mapping and risk prediction, there is a significant gap in the integration of spatially explicit machine learning models to predict and classify the co-endemicity of multiple NTDs with demographic and WASH variables. There remains limited research on using

machine learning techniques to model co-endemic transmission risks of STH, SCH, and LF in a unified framework that supports data-driven approaches while targeting interventions especially in Kenya. By leveraging Kenya's 2022 co-endemicity dataset from the ESPEN portal, this study provides actionable insights for optimizing NTDS co-endemicity and control programs.

The main objectives of this study were to develop machine learning models (RF, GBM, and XGBoost) to predict high STH prevalence and the co-endemicity status with LF and STH using demographic and WASH indicators, identify the key drivers of co-endemicity transmission and identify the high burden hotspots for targeted interventions.

## 2 Materials and methods

### 2.1 Study area

The study focused on the co-endemicity status of LF, STH and SCH in Kenya in 2022. All the regions required preventive chemotherapy.

### 2.2 Population description

In this study, secondary data analysis using the Expanded Special Project for Elimination of Neglected Tropical Diseases (ESPEN) in Kenya's co-endemicity data for the year 2022 was done. Kenya, on the east coast of Africa, is endemic for LF, Schistosomiasis and STH. The Ministry of Health implements a Strategic Plan for control of NTDs through the National NTD control program. This study analyzed data at both national and subnational levels (implementation units) including a total of 47 counties and 290 implementation units which was sufficient for modelling.

### 2.3 Data source and cleaning

The data used in this study is publicly available via the link https://espen.afro.who.int/maps-data/countries/kenya#overview--coendemicity. Electronic data capture, with standardized data collection forms, and automatic encrypted data upload to a secure central server. Form testing with specialists and stakeholders is responsible for conducting the surveys to ensure the form will collect quality data and adapted to the specificities of the diseases. Online dashboard setup for daily data visualization and near-real-time data cleaning using automated algorithms is available in the ESPEN portal. Each country owns their data and has their own restrictions. In this study, 2022 Kenyan dataset was used.

Data cleaning was carried out in R software version 4.5.0. The dataset contained 37 variables which were exclusively selected using the variable importance criterion. The missing values for the diseases data were detected during cleaning and were replaced using the mean of the complete samples. Since the data was balanced (35:65), there was no need of re-sampling the minority class. The data was balanced to a moderate extent to perform the training process to build the co-endemicity prediction. Missing values were replaced using mean imputation because the proportion of missing observations was relatively small and this approach allows preservation of the dataset size without introducing additional model complexity. Mean imputation is commonly used in exploratory machine learning analyses where missingness is minimal.

Sections 2.4 and 2.5 give more details on the variables used in the study.

**Flowchart of the proposed models and methodology.** The workflow for this research is illustrated in Fig 1 below where each process to be followed in outlined and linked to other subsequent processes. The workflow summarizes the data analysis processes the study will follow from the beginning to the end.

### 2.4 Outcome variable

The main target variable for this study was the co-endemicity status of the three NTDs (LF, STH and SCH) under consideration. Before the actual analysis using the machine learning algorithms, exploratory data analysis was conducted

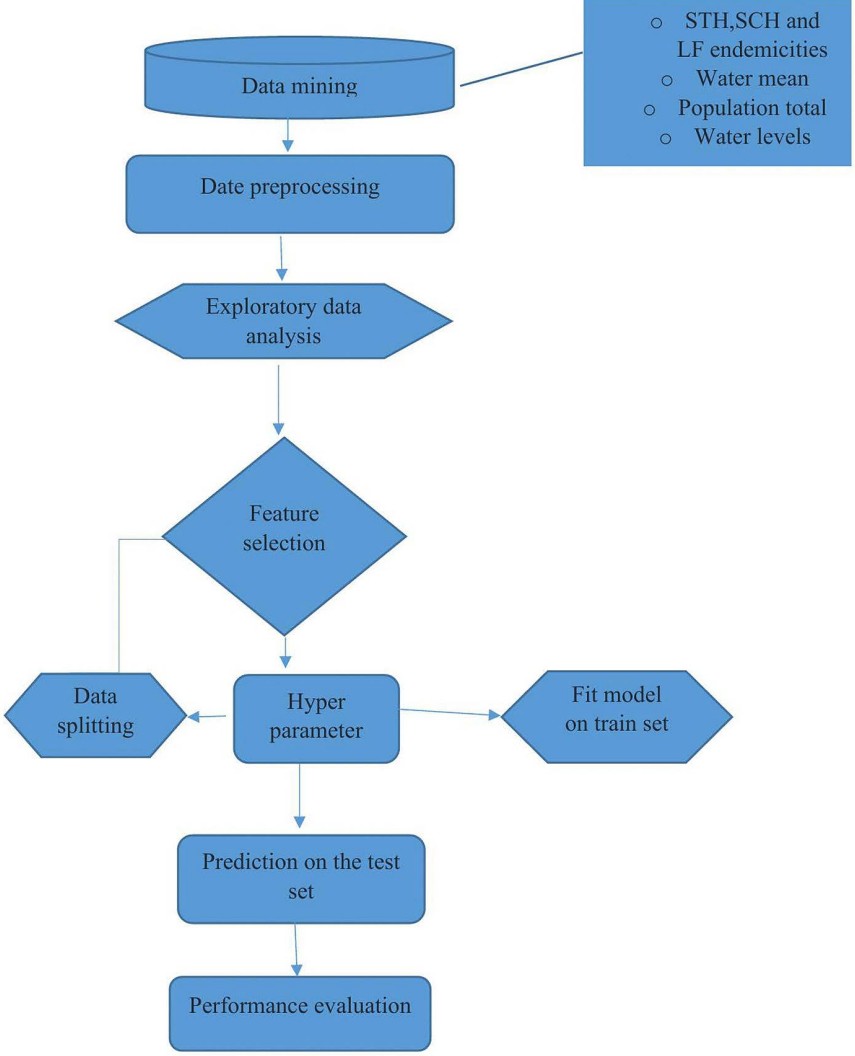

**Fig 1. Flowchart for the proposed methodology.**

to understand the transmission dynamics, patterns and the spatial distribution of the diseases. The raw data was transformed in order to be suitable for the machine learning methodology. Table 1 shows the key steps followed were the creation target variable, (co-endemicity class) whereby the disease endemicity status were combined into one categorical variable. One hot encoding was employed to convert categorical variables into binary outcomes for the machine learning compatibility.

## 2.5 Predictor variables

The predictor variables used in the study were high STH (categorical variable), population total (continuous variable), improved water means (continuous variable), improved sanitation means (continuous variable), LF, SCH and STH endemicities (categorical variable), water levels (continuous variable) and the endemicity IDs (continuous variable). Comprehensive analysis of all the predictor variables is shown in the results section (3.1).

**Table 1. Creation of target variable. As stepwise procedure of how the target variable was created.**

| LF + STH + SCH | Co - endemic status of all the diseases |
|---|---|
| LF + STH | Co - endemic for LF and STH only |
| LF + SCH | Co - endemic for LF and SCH only |
| STH + SCH | Co - endemic for STH and SCH only |
| LF only, STH only, SCH only | Endemic for only one disease |
| Non endemic | No disease endemicity |

## 2.6 Statistical analysis

Statistical analysis was done using the latest version of R (4.5.0). The relevant R libraries (XGBoost, RF, pROC, GBM) were installed prior to the analysis. Correlation and spatial analysis were conducted to give more insights into the data. Data splitting was done in the ratio 80:20 for the training and testing roles. Data splitting is important in machine learning classifications and methodologies as it ensures model evaluation on unseen data, data structuring and handling, and avoids leakage before preprocessing. Model performance metrics and confusion matrix were used to evaluate the models. Confusion matrix shows the true versus the predicted classes. To identify the most significant predictors, variable importance analysis was fitted for each of the machine learning models.

## 2.7 Machine learning methodology

**2.7.1 XGBoost algorithm.** The XGBoost model was used for enhanced prediction. The data was transformed to enhance the model fitting. The learning rate was 0.1 and the max_depth (tree depth) was set at 6. Early stopping was considered to prevent the model from over fitting and halting the training process in case the validation performance levels up. Feature importance by the XGBoost measures how each of the selected features contributes to the prediction process. The XGBoost algorithm handles the imbalanced data quite well as well as the large dataset.

**2.7.2 RF algorithm.** RF is a supervised ML algorithm that builds multiple decision trees and merges their predictions to improve accuracy and control over fitting. In our study, the RF algorithm was used for both prediction and classification tasks. It combines predictions through classification (majority vote (for your High/Low endemicity classes) and regression (average prediction). ROC was used to select the optimal model using the largest value. The final value used for the model was mtry = 2.

**2.7.3 Gradient boosting machine (GBM).** GBM is a sequential ensemble method that builds decision trees one at a time (unlike RF's parallel approach). Each new tree corrects errors of the previous ensemble and optimizes a loss function (e.g., deviance for classification) using gradient descent. The common tuning hyper parameters used in the current study includes an interaction. Depth = c (3, 5), n.trees = c (100, 200), shrinkage = c (0.01, 0.1) and n.minobsinnode = 10. Smaller shrinkage (≤ 0.1) with more n.trees (≥500) often performs better.

## 2.8 Algorithm selection justification

The study chose only three machine learning algorithms namely RF, GBM and XGBoost. Firstly, the three algorithms are tree-based ensemble methods known for robustness to non-linear and complex relationships which are common in disease data, handling mixed data types, i.e., categorical, continuous, discrete and without extensive feature scaling and lastly providing feature importance Scores (key for interpretability in disease modelling context). The Random Forest and XGBoost are the leading tree-based ensembles selected for their well-documented performance on structured data and their ability to capture non-linearities. Hyperparameter tuning was performed using cross-validation to identify optimal model parameters. The final models were then trained using the selected parameters and evaluated on the test dataset.

## 3. Results

### 3.1 Exploratory data analysis

Fig 2 below shows the distribtion of the population at risk of the common NTDs in Kenya. The Y axis shows the regions while the X axis represents the population totals. It gives insights on the distribution and geographical loaction of the diseases under study. The Rift valley region had the highest population which is at risk of contacting the diseases. Nairobi and North eastern have the lowest population. Nairobi, being the capital city might have improved water and access to better sanitation services. North Eastern regions is relatively dry with little or no rainfall [24], hence the risk of disease transmission is low.

As shown in Table 2, the grouping of diseases into co-endemicity categories was guided by public health intervention strategies used in preventive chemotherapy programs. Many NTD control programs implement integrated mass drug administration where treatment packages target multiple infections simultaneously. Therefore, modeling combinations of LF, SCH, and STH provides a more realistic representation of intervention planning and resource allocation.

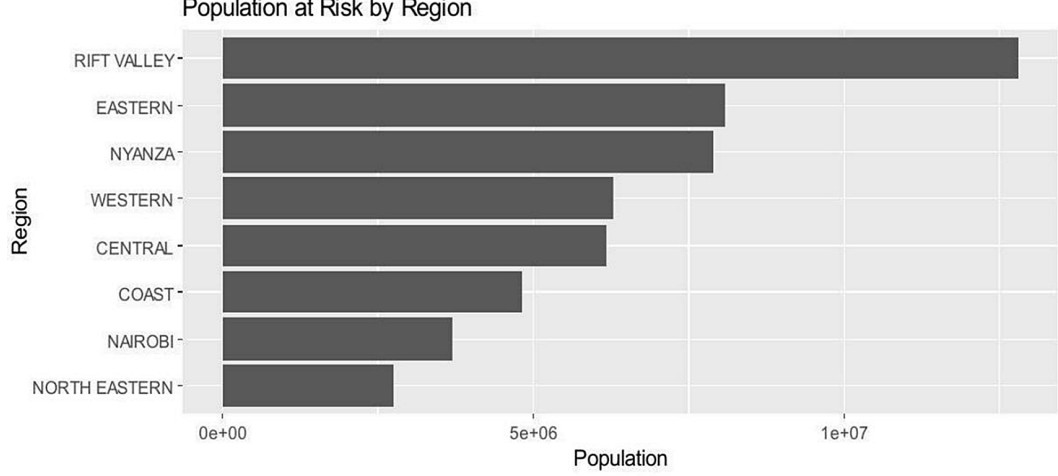

**Fig 2. The population at risk of NTDs by region.**

**Table 2. Frequency distribution of co-endemicity classes.**

| Co-endemicity class | Frequency | Percentage |
|---|---|---|
| LF + STH + SCH | 2 | 0.69 |
| LF + STH | 3 | 1.03 |
| LF + SCH | 3 | 1.03 |
| STH + SCH | 16 | 5.52 |
| Single disease | 119 | 41.03 |
| Non endemic | 147 | 50.69 |

The non-endemic areas dominate (50.69%), meaning more than half of the implementation units have none of the three diseases. Single disease endemicity (41.03%) is also common, indicating that many areas face only one of the diseases. Dual co-endemicity is relatively low (STH + SCH is the most frequent combination (5.52%)). Triple co-endemicity (LF + STH + SCH) is very rare (0.69%), suggesting limited overlap of all three diseases.

Fig 3 shows the STH endemicity levels in Kenya. The STH endmicity level is categorized into high prevalence, low prevalence, moderate prevalence and non endemic. The low and moderate prevalences are the commonest in STH transmission. The Y axis shows the number of geographical units that falls under each category.

Figs 4 and 5 shows the distribution of SCH and LF endemicity levels respectively. LF is sparsely distributed while the SCH distribution is high in all the four categories (High, low, moderate and non-endemic). The prevalence levels are categorized as high prevalence (>50%), moderate prevalence (20–49%), low prevalence (<20%) and non-endemic (areas where transmission did not occur).

Fig 6 shows the co-endemicity patterns by region. The coastal region has high LF and STH co-endemicity while western region has high SCH and STH overlap. The North Eastern region has few disease hotspots. The effectiveness of the MDA by region is shown in Fig 7. T2 is the commonest in majority of the regions. T1 works best in high co-endemicity regions.

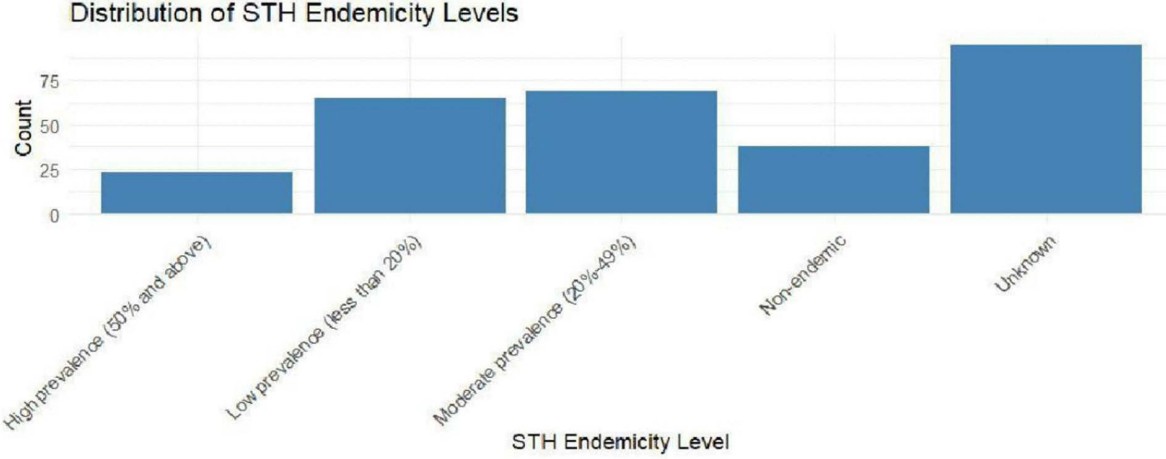

**Fig 3. Distribution of STH endemicity levels.**

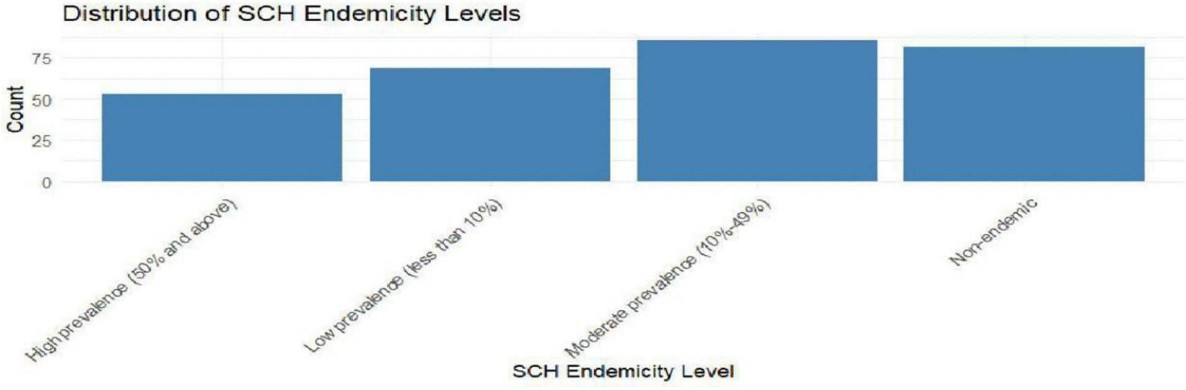

**Fig 4. Distribution of SCH endemicity levels.**

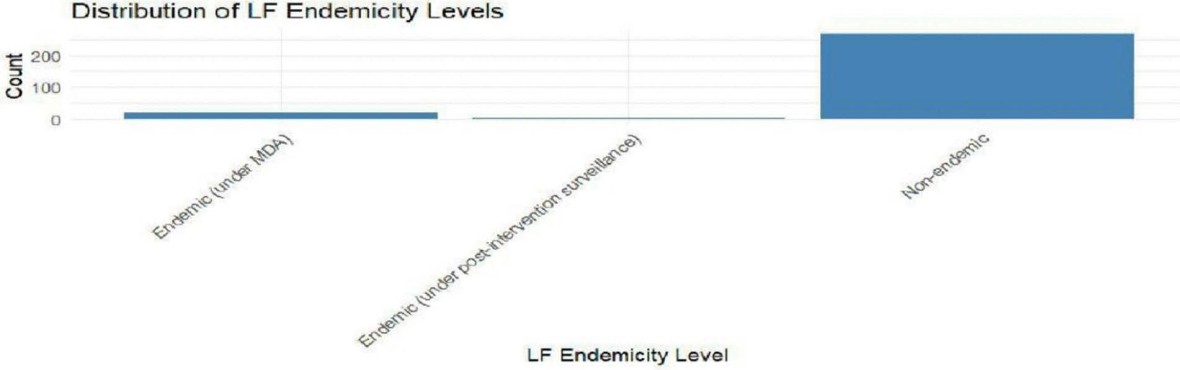

**Fig 5. Distribution of LF endemicity levels.**

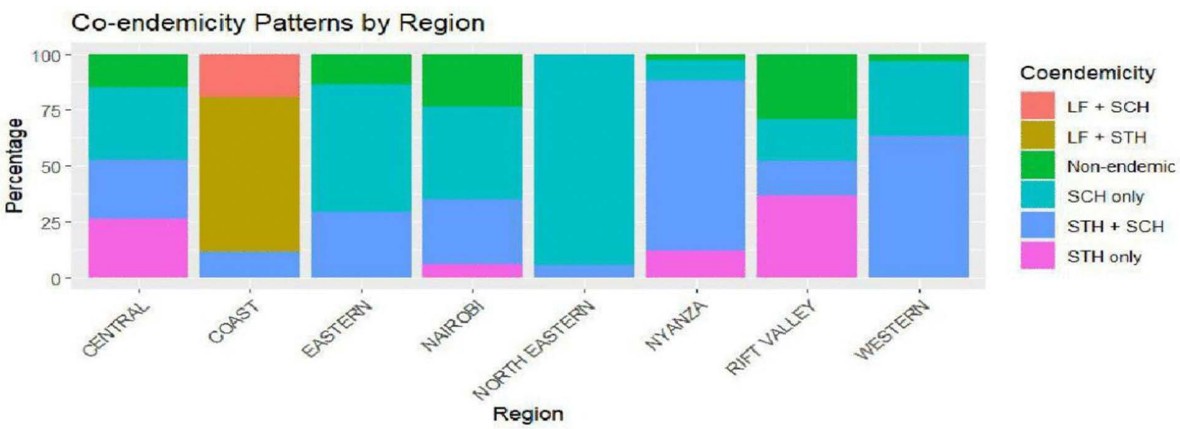

**Fig 6. NTDs co-endemicity status by region.**

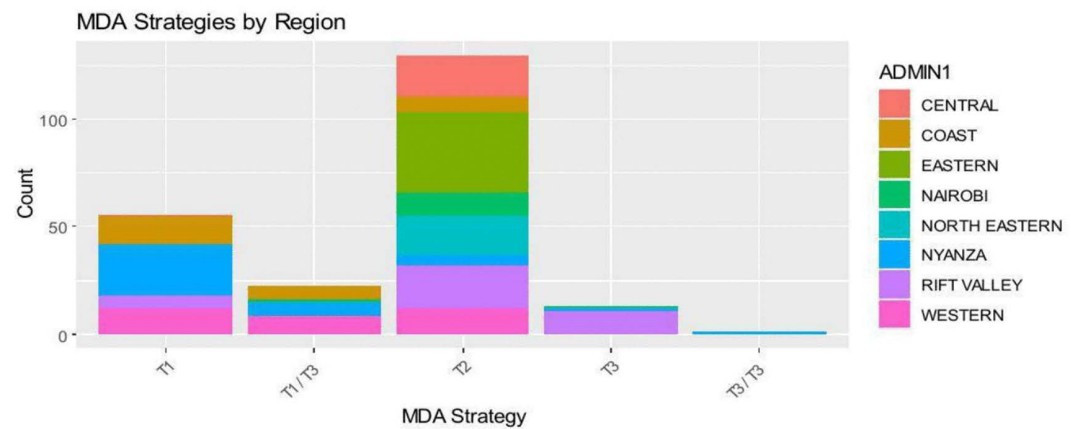

**Fig 7. MDA strategies and usage by region.**

## 3.2 Correlation between the variables

Table 3 shows the correlation analysis of the relationships between the predictor variables (water and sanitation means, endemicity levels and the population total). Pearson's (*r*) and the spearman's (*ρ*) were used since the data was a mixture of both continuous and categorical variables.

The correlation matrix in Table 3 unveils some important relationships between the disease endemicity status, population structure and the WASH variables. The STH has a weak negative correlation with the population of the region and a very weak negative correlation with the water access. Moreover, the STH is positively correlated with the improved water sanitation. LF endemicity is positively correlated with water access but negatively correlated with sanitation. The SCH endemicity has a weak positive correlation with the population and positively correlated with all the other variables. The WASH variables, (water and sanitation) are positively correlated with other. Access to improved water shows a positive correlation with the diseases as compared to sanitation.

The positive correlation observed between improved water access and disease prevalence may appear counter-intuitive. However, this relationship may reflect ecological and reporting biases. For example, regions with improved infrastructure often have stronger surveillance systems, resulting in higher detection rates of infections. Additionally, urban aggregation effects and population density may increase exposure opportunities despite improved water access. It is important to note that correlation analysis identifies statistical associations between variables but does not imply causal relationships. Therefore, the observed correlations should be interpreted cautiously and in conjunction with epidemiological evidence

## 3.3 Spatial analysis of STH prevalence

Fig 8 shows the spatial analysis of STH prevalence across the regions. Eastern and North eastern regions have the highest STH endemicity while Rift valley, Nyanza, coast and Nairobi regions have the lowest. These regions are highly favored and have better access to sanitation, environmental exposure and improved health programs.

## 3.4 The RF model

The RF model was trained to classify areas as "High" or "Low" disease risk based on 7 predictors using 233 samples. The model's classification results are as shown in Tables 4 and 5, Figs 8–10. ROC was used to select the optimal model using the largest value and the final value used for the model mtry (number of variables randomly sampled at each split) was 2. The class resampling cross validation was set at 5-fold.

Table 4 indicates that in the overall model, the predictive power was low since the ROC (0.5979) < 0.7. The sensitivity (0.6600) clearly indicates that only 66% of high-risk areas will be correctly identified and 53% (specificity) of low-risk areas will be misclassified as high-risk.

Table 3. Correlation matrix. The correlation matrix shows the relationship between the target variable and the predictors.

| Variable | Pop Tot | STH Endemicity | Water mean | Sanitation mean | LF Endemicity | SCH Endemicity |
|---|---|---|---|---|---|---|
| PotTot | 1.0000 | -0.0183 | 0.3344 | 0.1752 | 0.0042 | 0.0334 |
| STH Endemicity | -0.0183 | 1.0000 | -0.0282 | 0.1234 | -0.0590 | 0.0756 |
| Water mean | 0.3344 | -0.0282 | 1.0000 | 0.3526 | 0.1623 | 0.1612 |
| Sanitation mean | 0.1752 | 0.1234 | 0.3526 | 1.0000 | -0.1668 | 0.1301 |
| LF Endemicity | 0.0042 | -0.0590 | 0.1623 | -0.1668 | 1.0000 | 0.1607 |
| SCH Endemicity | 0.0334 | 0.0756 | 0.1612 | 0.1301 | 0.1607 | 1.0000 |

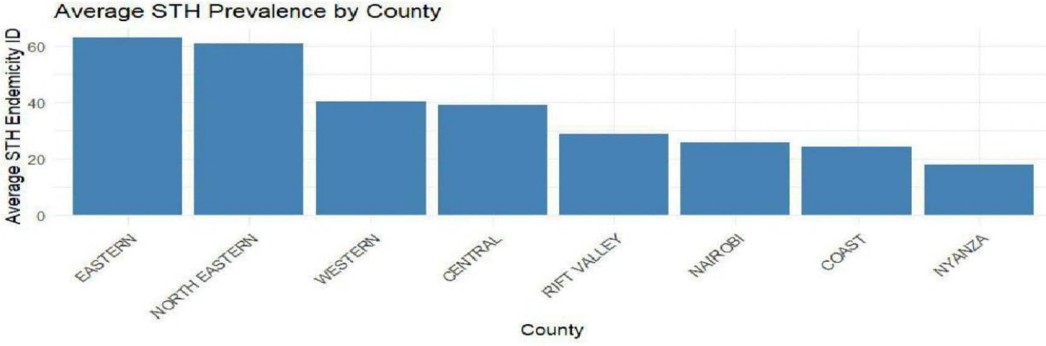

**Fig 8. STH prevalence by region in Kenya.**

**Table 4. Model overall performance metrics and statistics of the RF model.**

| Mtry | ROC | Sensitivity | Specificity |
|------|--------|------------|-------------|
| 2 | 0.5979 | 0.6600 | 0.4688 |
| 3 | 0.5851 | 0.6043 | 0.5346 |
| 4 | 0.5728 | 0.6040 | 0.5061 |
| 5 | 0.5511 | 0.5800 | 0.5061 |
| 7 | 0.5401 | 0.5800 | 0.4589 |

**Table 5. RF model key performance metrics.**

| Metric | Value |
|--------|-------|
| Sensitivity | 0.8387 |
| Accuracy | 0.6842 |
| Specificity | 0.5000 |
| Positive predicted | 0.6667 |
| Negative Predicted | 0.7222 |
| Prevalence | 0.5439 |
| Detection rate | 0.4561 |
| Detection prevalence | 0.6842 |
| Balanced accuracy | 0.6694 |
| Kappa | 0.3473 |
| No information rate (NIR) | 0.5439 |
| P value | 0.0219 |
| 95% CI | 0.5476, 0.8009 |

The RF model was plotted and shows the cross validation versus the number of predictors used as shown in Fig 9. The performance peaks at (0.59 – 0.60) predictive power and decreases across the predictors.

**3.4.1 Variable importance features of the RF model.** In Fig 10, the variable importance graph shows that the population of the region was the most important predictor while the water level (high low) was the least.

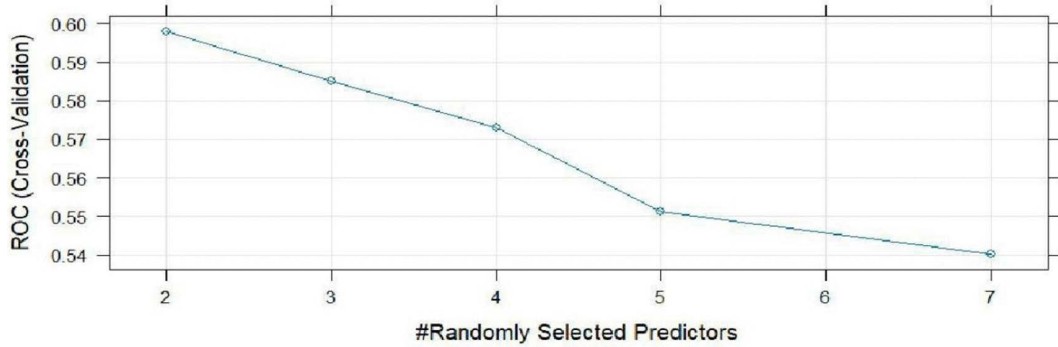

**Fig 9. Random Forest plot.**

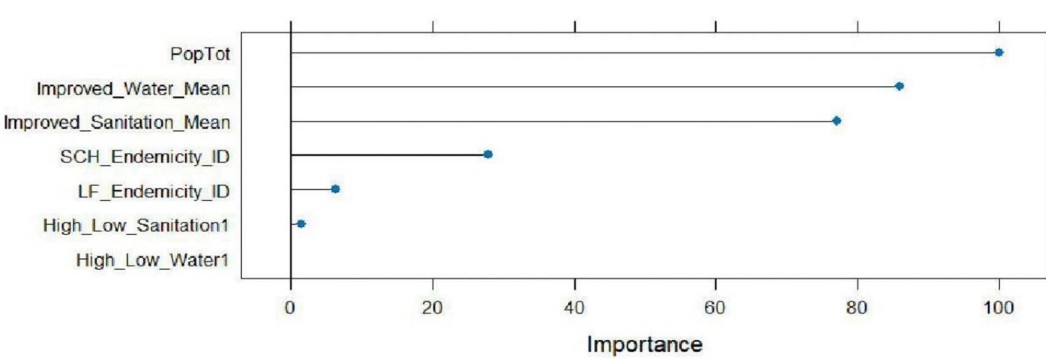

**Fig 10. RF Variable importance.**

**3.4.2 Confusion matrix performance.** A confusion matrix is a table used to evaluate the performance of a classification machine learning model. It shows how well your model's predicted classifications match the actual labels. The resulting confusion matrix for the RF model is shown in equation (1).

$$\begin{pmatrix} 26(TP) & 13(FP) \\ 5(FN) & 13(TN) \end{pmatrix}$$

(1)

The model correctly predicted 26 (TP) and 13 (TN) accounting for 68.42% (positive predicted) accuracy where TP (True positives), FP (False positive), FN (False Negative) and TN (True Negatives).

Table 5 shows that in the prediction task, the RF model accurately predicted 68.42% of the disease's cases. The wide confidence interval suggests a good variability in the model. The P - value (0.0219) was statistically significant and was better than random guessing (NIR = 54.39%). The model's sensitivity (83.87%) means that it was able to detect the actual cases and was able to minimize false negatives. Only 50% of the actual low cases were correctly identified (specificity). The mean of sensitivity and specificity (balanced accuracy was 0.6694). The Kappa, which is interpreted as a fair agreement between the variables was 0.3473.

Fig 11 shows the ROC curve of the RF model and it indicates that it had moderate predictive power (AUC = 0.703). A ROC curve is a graphical representation of a classification model's performance at all possible classification thresholds. It helps you evaluate the trade-off between sensitivity and specificity.

Fig 11 shows the receiver operating characteristic, area under the curve (usually a plot of sensitivity against specificity) for the RF model.

## 3.5 The GBM model

GBM is a powerful ensemble learning algorithm model known for its strong predictive performance, especially in tabular data [25]. The resampling was set at 5 – fold cross validation with 7 predictors, 233 samples and 2 binary classes. The optimal hyper parameters were; n.trees = 50, interaction.depth = 2, shrinkage = 0.1, and n.minobsinnode = 10. The tuning parameter 'shrinkage' was held constant at a value of 0.1. Tuning parameter 'n.minobsinnode' was held constant at a value of 10. The ROC was used to select the optimal model using the largest value. The GBM model results are as shown in Figs 12–14; Tables 5 and 6.

Fig 12 shows the results from the GBM hyper parameters. There is a competing performance from depths 1–3 with no clear winning case. There is an observable underperformance in the current model as compared to the RF.

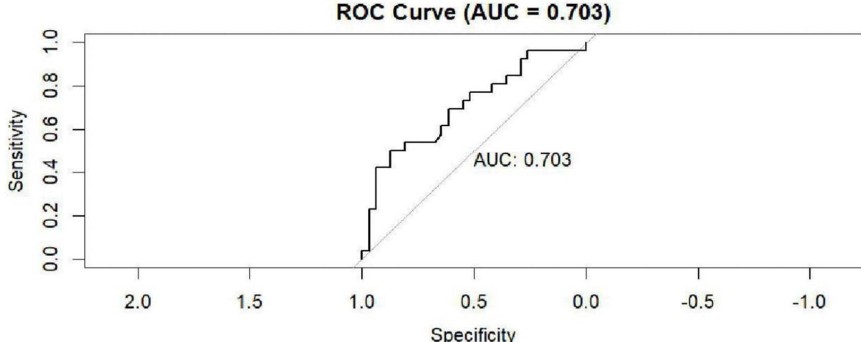

**Fig 11. RF ROC AUC.**

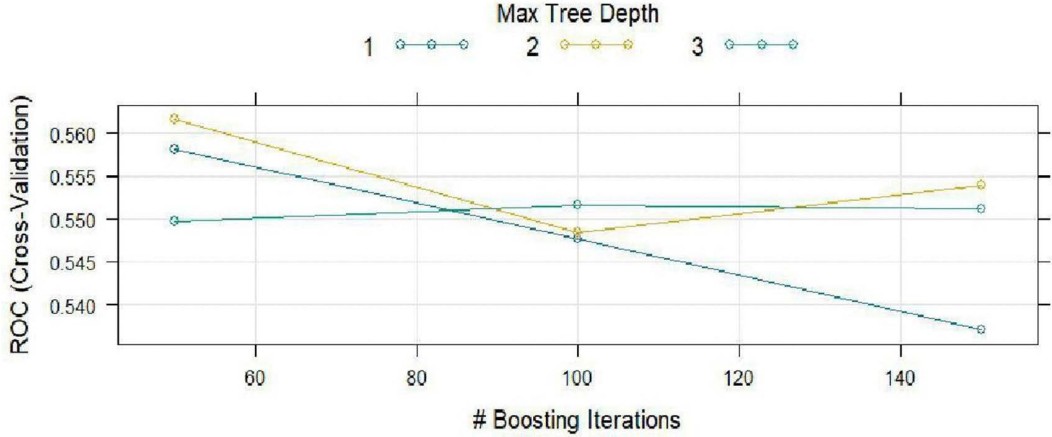

**Fig 12. GBM model hyper parameter results.**

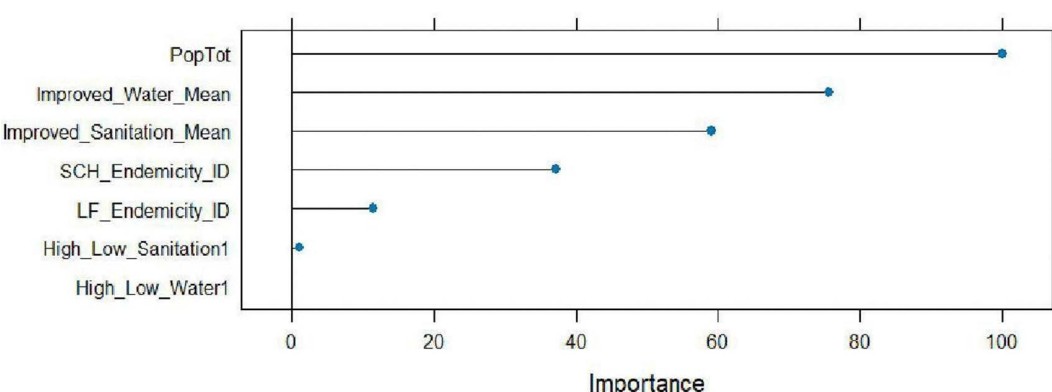

**Fig 13. GBM variable importance.**

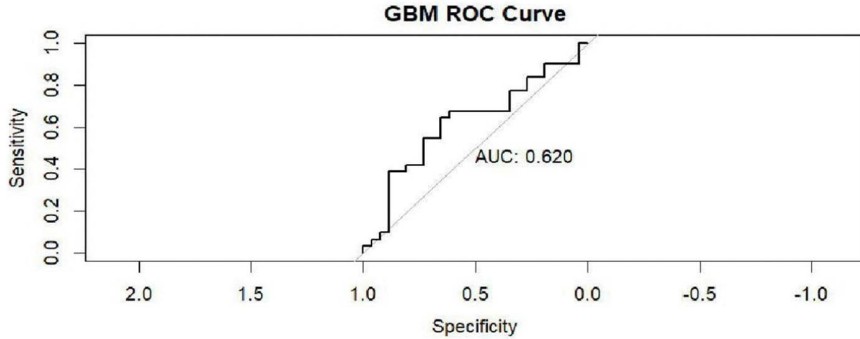

**Fig 14. ROC curve of the GBM model.**

**Table 6. Resampling results across tuning parameters for the GBM model.**

| Interaction depth | n.trees | ROC | Sensitivity | Specificity |
|---|---|---|---|---|
| 1 | 50 | 0.4922 | 0.7068 | 0.3065 |
| 1 | 100 | 0.5444 | 0.6745 | 0.3723 |
| 1 | 150 | 0.5526 | 0.6748 | 0.4286 |
| 2 | 50 | 0.5230 | 0.6111 | 0.4273 |
| 2 | 100 | 0.5603 | 0.6108 | 0.4931 |
| 2 | 150 | 0.5691 | 0.6108 | 0.4641 |
| 3 | 50 | 0.5455 | 0.6114 | 0.3987 |
| 3 | 100 | 0.5598 | 0.6108 | 0.4732 |
| 3 | 150 | 0.5540 | 0.5945 | 0.5208 |

Fig 12 shows the hyper parameter results of the GBM model comprising of the maximum tree depths of different boosting iterations.

Table 6 shows the results of resampling across the tuning parameters. The best model was identified using the ROC metric (0.5691) and had ***n.trees*** = **150** and a depth of 2. The highest sensitivity (0.7068) was observed at interaction depth 1 and had a specificity of 0.3065. The final values used for the model were n.trees = 50, interaction.depth = 2, shrinkage = 0.1 and n.minobsinnode = 10.

**3.5.1 GBM variable importance.** Fig 13 shows the variable importance of the GBM model. The total population, improved water and sanitation were identified as the most important predictors.

**3.5.2 GBM performance on the test set.** In Table 7, the model correctly predicted 61.4% of the cases. The *p*–value (0.1762) was not significant as compared to the NIR (54.39%). The model is good at identifying high cases (sensitivity = 77.42%). The kappa (0.2023), showed a slight agreement with the results. The model performed poorly in identifying low cases (specificity = 42.31%). A graph of sensitivity versus specificity quantifying the area under the ROC curve is shown in Fig 14.

**3.5.3 Confusion matrix of the GBM model.** In equation (2), the model correctly predicted 24 (TP) and 11 (TN) accounting for 61.40% (positive predicted) accuracy where TP (True positives), FP (False positive), FN (False Negative) and TN (True Negatives).

$$\begin{pmatrix} 24(TP) & 15(FN) \\ 7(FP) & 11(TN) \end{pmatrix}$$

(2)

Fig 14 reveals that the GBM model achieved an AUC of 0.620, indicating moderate classification performance. While this score is above the threshold of random guessing (AUC = 0.5), it suggests that the model's ability to distinguish between classes is modest and could benefit from further optimization or additional features.

## 3.6 The XGBoost model

The XGBoost model was evaluated using the tuning hyper parameters, ROC curve, variable importance features and the confusion matrix. The tuning parameter, $\gamma$ ' was held constant at a value of 0. Tuning parameter 'min_child_weight' was held constant at a value of 1. ROC was used to select the optimal model using the largest value. The final values used

**Table 7. Performance metrics of the GBM model.**

| Metric | Value |
|---|---|
| Accuracy | 0.6140 |
| 95% CI | 0.4757, 0.74 |
| No Information Rate (NIR) | 0.5439 |
| P value | 0.1762 |
| Kappa | 0.2023 |
| Sensitivity | 0.7742 |
| Specificity | 0.4231 |
| Positive predicted | 0.6154 |
| Negative predicted | 0.6111 |
| Prevalence | 0.5439 |
| Detection rate | 0.4211 |
| Detection prevalence | 0.6842 |
| Balanced accuracy | 0.5986 |

for the model were nrounds = 100, max_depth = 1, eta = 0.4, gamma = 0, colsample_bytree = 0.6, min_child_weight = 1 and subsample = 0.875. The key findings and metrics of the XGBoost model are as indicated in Figs 14–16; Table 8. The confusion matrix is shown in equation (3)

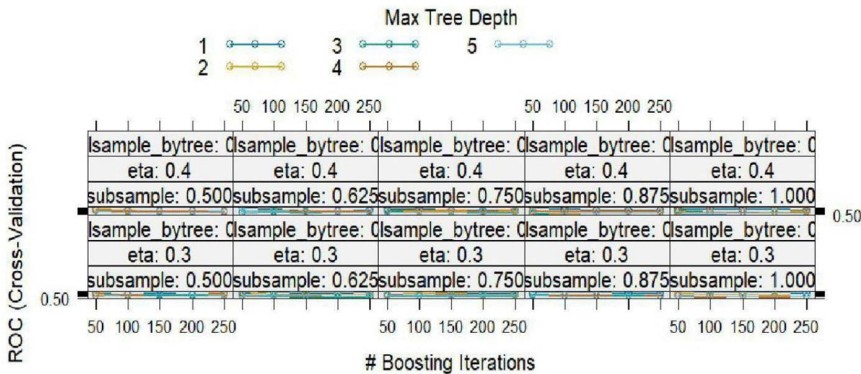

**Fig 15. XGBoost model illustration.**

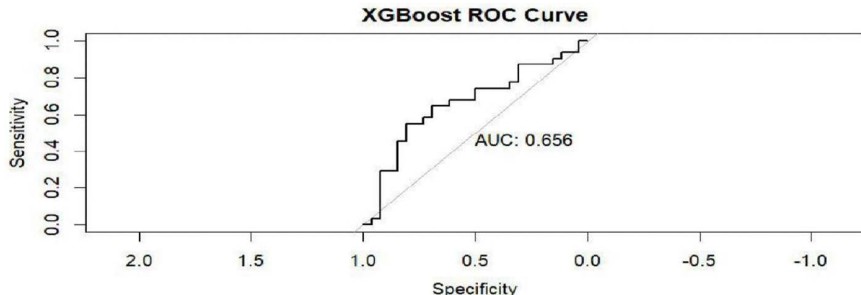

**Fig 16. XGBoost ROC curve.**

**Table 8. XGBoost performance metrics.**

| Metric | Value |
|---|---|
| Accuracy | 0.5965 |
| 95% CI | 0.4582,0.7244 |
| NIR | 0.5439 |
| P value | 0.2540 |
| Kappa | 0.1687 |
| Sensitivity | 0.7419 |
| Specificity | 0.4231 |
| Positive predicted | 0.6053 |
| Negative predicted | 0.5789 |
| Prevalence | 0.5439 |
| Detection rate | 0.4035 |
| Detection prevalence | 0.6667 |
| Balanced accuracy | 0.5825 |

PLOS Neglected Tropical Diseases

### 3.6.1 XGBoost model illustration.
XGBoost is a powerful, scalable machine learning algorithm based on gradient boosting. It builds an ensemble of decision trees sequentially, where each new tree tries to correct the errors of the previous ones [26]. Fig 15 shows the illustration of XGBoost model with the decision trees using hyper parameters.

### 3.6.2 XGBoost model ROC curve.
In Fig 16, the XGBoost model attained an AUC of 0.6560. The XGBoost model ROC curve was used to evaluate the relationship between the true positive rates (sensitivity) and the false positive rate (specificity).

### 3.6.3 XGBoost variable importance features.
The feature importance technique was applied in the XGBoost model to rank how much each feature (input variable) contributed to the predictive power of a model. As shown in Fig 17, the improved water mean (WASH variable) and the population total were the most important variables in predicting the disease co-endemicity under the XGBoost model.

### 3.6.4 Confusion matrix and metrics.
The confusion matrix of the XGBoost model is shown in equation (3) and the overall performance metrics are shown in Table 8.

$$\begin{pmatrix} 23(TP) & 15(FN) \\ 8(FP) & 11(TN) \end{pmatrix}$$

(3)

As shown in Table 8, the model's p-value (0.2540) shows that it isn't significant in the predictions. Based on the overall performance, the sensitivity (74.19%), specificity (42.31%) and accuracy (59.65%), shows that the model is reliable in detecting high cases, poor at identifying low cases and moderate in the prediction task respectively.

### 3.6.5 Models' comparison.
A comprehensive comparison of all the three models (RF, GBM and XGBoost) shown in Fig 18. The RF model (AUC = 0.703) emerged the highest among the three while GBM model (AUC = 0.62) was the lowest. The XGBoost (AUC = O.656) model was the middle in performance.

In Table 9, the models performance statistics based on ROC curves are shown. The RF model was the strongest with a maximum AUC of 0.775 and a mean of 0.5979. The XGBoost unveiled a competitive mean of 0.6167.

### 3.6.6. Overall Models comparison and performance.
An overall performance of all the models based on the key performance metrics (Accuracy, Sensitivity, specificity, kappa and 95% confidence interval is shown in Table 10.

Table 10 shows an overall performance of all the models based on the key performance metrics (Accuracy, Sensitivity, specificity, kappa and 95% confidence interval).

**Fig 17. XGBoost variables importance.**

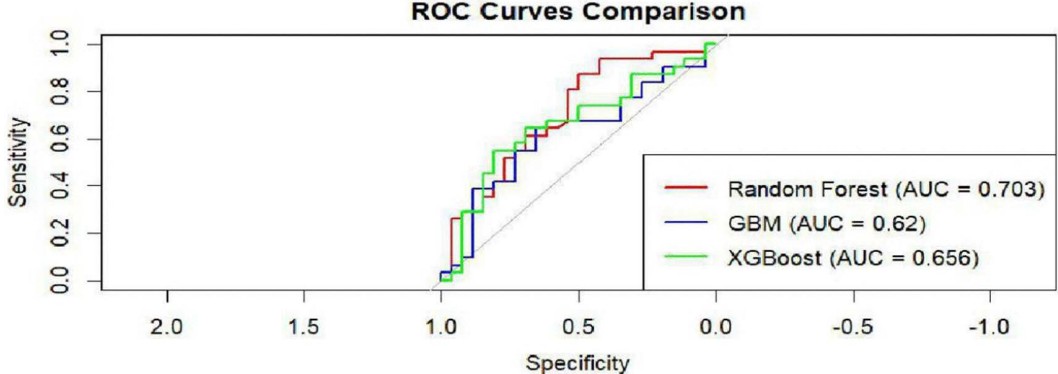

**Fig 18. Models comparison.**

**Table 9. Models performance based on the ROC curves.**

| ROC | Min | Median | Mean | Maximum |
|---|---|---|---|---|
| RF | 0.5128 | 0.5582 | 0.5979 | 0.7752 |
| GBM | 0.4809 | 0.5627 | 0.5616 | 0.6038 |
| XGBoost | 0.5273 | 0.6026 | 0.6167 | 0.6857 |

**Table 10. Models performance based on selected metrics.**

| Metric | RF | GBM | XGBoost |
|---|---|---|---|
| ROC AUC | 0.7030 | 0.6200 | 0.6560 |
| Accuracy | 0.6842 | 0.6140 | 0.5965 |
| Sensitivity | 0.8387 | 0.7742 | 0.7419 |
| Specificity | 0.5000 | 0.4231 | 0.4231 |
| Kappa | 0.3473 | 0.2023 | 0.1687 |
| 95% CI | 0.5476,0.8009 | 0.4757,0.7400 | 0.4582,0.7244 |

With a value of 0.7030, RF obtained the highest Area under the Receiver Operating Characteristic Curve (ROC AUC), demonstrating the best ability to distinguish between the positive and negative classes. Significantly better than the performance of the other two models (Kappa of 0.2023 for GBM and 0.1687 for XGBoost), this superior discrimination translated into a noteworthy Accuracy of 0.6842 and a Kappa statistic of 0.3473, suggesting moderate agreement beyond chance. With ROC AUC values of 0.6200 and 0.6560, respectively, the GBM and XGBoost models performed similarly, falling short of the RF approach's predictive power.

## 4. Discussion

The Expanded Special Project for the Elimination of Neglected Tropical Diseases (ESPEN) is collaborating directly with the ministries and other stakeholders of the Neglected Tropical Diseases (NTDs) to amplify the impact and elimination initiatives. In Africa, the total population which required preventive chemotherapy reduced from 592 million in 2016–588 million in 2019. This positive response was as a result of surveillance and improvement in research. Two countries namely Togo and Malawi were validated for completely eliminating lymphatic filariasis (LF) in 2017 and 2020 respectively. In

Kenya, elimination and control of NTDs has been prioritized. The NTDs disproportionately affect the most vulnerable, marginalized people in the poorest, most remote communities around the world.

The exploratory data analysis showed that all the Kenya's regions were at risk of NTDs. The endemicty levels of the three diseases (STH, LF and SCH) showed moderate to high prevalence. In terms of distribution and area coverage, the STH and SCH had higher distribution levels as compared to LF.

Our analysis demonstrates that machine learning models, particularly the RF model, effectively predict high STH prevalence areas in Kenya (AUC = 0.7030). The results of the RF model emerging as the most powerful for NTDs prediction and classification task aligns with those of [27–29]. Furthermore, RF algorithm is less prone to over fitting than the other models because it uses bootstrap aggregating (bagging) and random subspace sampling. It is less sensitive to hyper parameter tuning and its design inherently handles noise/outliers better. GBM and XGBoost, on the other hand, employ boosting, training trees one after the other, fixing the mistakes (residuals) of the previous tree. They are more prone to over fitting and capturing noise when applied to unseen (test) data, even though this reduces bias and frequently produces excellent performance on the training set. In contrast to GBM and particularly XGBoost, which frequently need extensive and meticulous hyper parameter tuning to prevent over fitting and attain optimal generalization performance, RF is frequently very effective with its default hyper parameter settings. The boosting models' propensity to over fit the training data was probably not adequately controlled by the default or basic tuning that was used.

The models identified improved sanitation, population total and water access as the strongest protective factors, consistent with global evidence linking WASH to STH reduction [19]. Notably, co-endemicity with LF and SCH emerged as significant predictors, suggesting overlapping transmission pathways that warrant integrated control strategies as evidenced in studies by [30,31].

Spatial analysis revealed persistent hotspots in Eastern and North eastern counties, aligning with historical prevalence patterns [32]. These regions also reported lower WASH coverage, reinforcing the need for infrastructure investments alongside MDA. Conversely, Central and Nairobi counties exhibited lower STH burden, likely due to better urban sanitation systems. The study findings further recommends that Sanitation campaigns in schools and communities could disrupt STH transmission more effectively than standalone deworming.

Our findings concur with studies linking poor WASH to STH risk [6] but extend prior work by quantifying variable importance using ML. For instance, improved sanitation outweighed population density, suggesting that WASH interventions may yield greater impact than demographic factors alone. This contrasts with earlier regression-based models that emphasized population density highlighting ML's ability to capture complex interactions.

Lastly, although the Random Forest model achieved the highest predictive performance (AUC = 0.703), the overall model performance can be considered moderate. Several factors may explain this outcome. First, the dataset consisted of a limited number of implementation units (n = 290) and a relatively small number of predictors, which may constrain the model's ability to capture complex transmission dynamics. Secondly, NTD transmission is influenced by multiple environmental, climatic, socioeconomic, and behavioral factors that were not fully captured in the available dataset. Third, the co-endemicity outcome variable combines multiple disease states into a single classification framework, which increases classification complexity and may reduce predictive discrimination.

The relatively low specificity observed across the models indicates that some low-risk areas may be incorrectly classified as high-risk. From a public health perspective, this may lead to over-allocation of resources to certain areas. However, in disease control programs, prioritizing sensitivity (detecting true high-risk areas) is often preferred because failing to identify high-transmission regions could undermine elimination efforts.

## 5. Conclusion

The study has demonstrated that machine learning methodology has the potential of classifying and predicting NTDs in Kenya with a key focus on the co-endemicity of STH, SCH and LF. Additionally, integration of the WASH variables

and population data with the machine learning models provided insights and knowledge on the risk factors and the main hotspots in Kenya. Furthermore, the key findings indicated that RF model and the XGBoost outperformed the GBM model in both the prediction and classification tasks. In the overall performance, the RF outperformed all the other models.

The exploratory analysis showed that the main co-endemic zones (Eastern and North eastern) have access to poor infrastructure and harsh climatic conditions hence underscoring the resistance of NTDs in those regions. In general, the study re-affirms that machine learning techniques are not only valuable tools for risk prediction but also a strategic enabler for targeted interventions, resource allocation, and proactive NTDs surveillance, especially in data-limited settings.

## 6. Recommendations

This study recommends the use of real-time machine learning dashboards for dynamic risk mapping and timely interventions. The study further recommends the cost-effectiveness analyses of water and sanitation versus mass drug administration strategies for maximum optimization. The Integration with climate data to forecast environmental transmission risks of the NTDs hence promoting one heath approach to disease control and mitigation measures. The high-risk counties should be prioritized for mass drug administration procedures and other health programs to curb disease severity and spread.

Future machine learning models should prioritize interpretability and explain ability to increase trust and adoption among policymakers, allowing them to clearly see how variables and risk factors drive NTDs risk. Training programs and collaborations with local universities and health ministries should be strengthened to build capacity in machine learning, artificial intelligence, geospatial and geo statistical approaches and evidence-based NTDs control.

## 7. Study limitations

The study provided important knowledge on the applications of machine learning in NTDs modelling, however, there were some notable limitations. Firstly, comprehensive data was not available and some areas were faced with comparability of prevalence. There were limited social and demographic co-variates which could influence disease transmissions. Another limitation relates to the moderate predictive performance of the models (AUC range: 0.62–0.70). While these values indicate predictive ability above random classification, they also suggest that some misclassification may occur when identifying high-risk areas. Therefore, model outputs should be interpreted as decision-support tools rather than definitive predictors of disease transmission. Mean imputation may introduce bias by underestimating variability in the dataset. Future work could explore more robust imputation approaches such as k-nearest neighbors (k-NN) or multiple imputation to improve data representation.

## Supporting information

**S1 Data. Co-endemicity dataset for neglected tropical diseases in Kenya (2022).** The dataset contains implementation-unit level data for 290 administrative areas in Kenya used to assess the co-endemicity of major neglected tropical diseases (NTDs), including Lymphatic Filariasis (LF), Soil-Transmitted Helminth Infections (STH), and Schistosomiasis (SCH). The dataset includes geographic identifiers, demographic information, disease endemicity status, and environmental indicators relevant to disease transmission and control.
(CSV)

## Acknowledgments

The authors extend their appreciation to their universities for supporting their research work.

## Author contributions

**Conceptualization:** Nkuba Nyerere, Damaris Felistus Mulwa.

**Data curation:** Nkuba Nyerere, Damaris Felistus Mulwa.

**Formal analysis:** Damaris Felistus Mulwa.

**Investigation:** Nkuba Nyerere.

**Methodology:** Nkuba Nyerere, Damaris Felistus Mulwa.

**Software:** Damaris Felistus Mulwa.

**Supervision:** Nkuba Nyerere.

**Validation:** Damaris Felistus Mulwa.

**Visualization:** Damaris Felistus Mulwa.

**Writing – original draft:** Nkuba Nyerere, Damaris Felistus Mulwa.

**Writing – review & editing:** Damaris Felistus Mulwa.

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
