## [Decision Letter · Decision Letter 0]

26 Aug 2025

Advanced Machine Learning Approaches for Predicting Neglected Tropical Disease (NTD) Co-endemicity in Kenya: A Focus on Soil-Transmitted Helminths, Schistosomiasis, and Lymphatic Filariasis.

Dear Dr. MULWA,

Thank you for submitting your manuscript to PLOS Neglected Tropical Diseases. After careful consideration, we feel that it has merit but does not fully meet PLOS Neglected Tropical Diseases's publication criteria as it currently stands. Therefore, we invite you to submit a revised version of the manuscript that addresses the points raised during the review process.

Please submit your revised manuscript within 60 days Oct 25 2025 11:59PM. If you will need more time than this to complete your revisions, please reply to this message or contact the journal office at plosntds@plos.org. Please include the following items when submitting your revised manuscript:

We look forward to receiving your revised manuscript.

Kind regards,

Anand Setty Balakrishnan, PhD

Academic Editor

Qu Cheng

Section Editor

Shaden Kamhawi

co-Editor-in-Chief

Paul Brindley

co-Editor-in-Chief

**Additional Editor Comments:**

Authors clarify the reviewers comments and submit the revised version for further decision

**Journal Requirements:**

1) Please provide an Author Summary. This should appear in your manuscript between the Abstract (if applicable) and the Introduction, and should be 150-200 words long. The aim should be to make your findings accessible to a wide audience that includes both scientists and non-scientists. Sample summaries can be found on our website under Submission Guidelines:

Potential Copyright Issues:

- Figure 1. Please (a) provide a direct link to the base layer of the map (i.e., the country or region border shape) and ensure this is also included in the figure legend; and (b) provide a link to the terms of use / license information for the base layer image or shapefile. We cannot publish proprietary or copyrighted maps (e.g. Google Maps, Mapquest) and the terms of use for your map base layer must be compatible with our CC BY 4.0 license.

**Reviewers' Comments:**

Reviewer's Responses to Questions

**Key Review Criteria Required for Acceptance?**

**Methods:**

-Are the objectives of the study clearly articulated with a clear testable hypothesis stated?

-Is the study design appropriate to address the stated objectives?

-Is the population clearly described and appropriate for the hypothesis being tested?

-Is the sample size sufficient to ensure adequate power to address the hypothesis being tested?

-Were correct statistical analysis used to support conclusions?

-Are there concerns about ethical or regulatory requirements being met?

Reviewer #1: 1) Model comparison was done only using AUC? Why not with other parameters?

2) The comparison graph AUC is not scaled properly and the graph can be much improved.

3) Why you have chosen only three ML algorithms namely RF, GBM and XGboost? Any specific reasons?

4) Why other classification algorithms are not chosen? Give Reasons.

5) What is the reason RF is giving good results compared to other algorithms take for experiment? Discuss.

6) Whether the classes in the dataset are uniformly distributed? Class im-balance is there in the dataset? No discussion on this.

7) How many features are in the dataset? Whether you considered all? No discussion in this regard.

8) No details about the size of the dataset?

9) Why deep learning models are not preferred for the given problem?

10) Details such as missing values, categorical variables and other details in exploratory data analysis are missing.

11) Figure 1 need to be of high clarity. Currently it is not so. Please change. Highlight the area which is being focused in this research.

12) How figure 2 is plotted? No details.

13) English throughout the manuscript need to be revisited.

14) Figures and the tables can be inlined along with the text. Why it is placed at the end of the paper?

15) Architecture about the proposed methodology may be drawn in the paper and placed in the methodology section.

The authors are suggested to give more detail about the dataset and clarify the questions asked above.

Reviewer #2: (No Response)

Reviewer #3: Population Description

Population = Kenya subnational units with co-endemicity data (LF, SCH, STH). This is clearly defined and appropriate for testing the hypothesis. This could be improved by adding: total number of districts/units analyzed, and justification for inclusion/exclusion.

How many geographic units were analyzed, and is this sufficient for model training/testing (especially with 80:20 split)?”

**Results:**

-Does the analysis presented match the analysis plan?

-Are the results clearly and completely presented?

-Are the figures (Tables, Images) of sufficient quality for clarity?

Reviewer #1: Figures and tables clarity, scaling and aesthetic sense has to be improved.

Reviewer #2: The results of the machine learning models are explained effectively for each model. I did not find the reasons why the RF model's performance is better than that of other models. The study done by the authors did not compare the results with the state of the art. The authors must address the requirement for cross-validation in their study.

Reviewer #3: include AUC values, sensitivity/specificity, kappa statistic to show robustness.

**Conclusions:**

-Are the conclusions supported by the data presented?

-Are the limitations of analysis clearly described?

-Do the authors discuss how these data can be helpful to advance our understanding of the topic under study?

-Is public health relevance addressed?

Reviewer #1: Drafted the conclusion, currently it is not impactful and highlight the future directions.

Reviewer #2: I did not find any issue in the conclusion. While incorporating the comments in the article, they can modify the conclusion and abstract.

Reviewer #3: he conclusion is strong and well-aligned with objectives, but can be tightened by adding quantitative results, softening causal claims, and briefly acknowledging limitations/future work

**Editorial and Data Presentation Modifications?**

Reviewer #1: Major revision

Reviewer #2: NIL

Reviewer #3: Include a summary table comparing performance metrics (accuracy, AUC, sensitivity, specificity, kappa) for each model.

Use variable importance plots to visually show key predictors (e.g., sanitation, water access, population size).

A map figure highlighting predicted hotspots (Eastern & Northeastern Kenya) strengthens spatial interpretation.

tighten language, avoid causal overstatements, add limitations/future work.

**Summary and General Comments**

Reviewer #1: 1) Model comparison was done only using AUC? Why not with other parameters?

2) The comparison graph AUC is not scaled properly and the graph can be much improved.

3) Why you have chosen only three ML algorithms namely RF, GBM and XGboost? Any specific reasons?

4) Why other classification algorithms are not chosen? Give Reasons.

5) What is the reason RF is giving good results compared to other algorithms take for experiment? Discuss.

6) Whether the classes in the dataset are uniformly distributed? Class im-balance is there in the dataset? No discussion on this.

7) How many features are in the dataset? Whether you considered all? No discussion in this regard.

8) No details about the size of the dataset?

9) Why deep learning models are not preferred for the given problem?

10) Details such as missing values, categorical variables and other details in exploratory data analysis are missing.

11) Figure 1 need to be of high clarity. Currently it is not so. Please change. Highlight the area which is being focused in this research.

12) How figure 2 is plotted? No details.

13) English throughout the manuscript need to be revisited.

14) Figures and the tables can be inlined along with the text. Why it is placed at the end of the paper?

15) Architecture about the proposed methodology may be drawn in the paper and placed in the methodology section.

The authors are suggested to give more detail about the dataset and clarify the questions asked above.

Reviewer #2: NIL

Reviewer #3: Avoid repetition of phrases like “outperformed” — state once with evidence.

Replace long connectors (“hence underscoring the resistance of NTDs in those regions”) with precise wording.

PLOS authors have the option to publish the peer review history of their article (what does this mean?). If published, this will include your full peer review and any attached files.). If published, this will include your full peer review and any attached files.). If published, this will include your full peer review and any attached files.). If published, this will include your full peer review and any attached files.

...

Reviewer #1: No

Reviewer #2: No

Reviewer #3: No

**Figure resubmission:**
---

## [Decision Letter · Decision Letter 1]

22 Feb 2026

Advanced Machine Learning Approaches for Predicting Neglected Tropical Disease (NTD) Co-endemicity in Kenya: A Focus on Soil-Transmitted Helminths, Schistosomiasis, and Lymphatic Filariasis.

Dear Dr. MULWA,

Thank you for submitting your manuscript to PLOS Neglected Tropical Diseases. After careful consideration, we feel that it has merit but does not fully meet PLOS Neglected Tropical Diseases's publication criteria as it currently stands. Therefore, we invite you to submit a revised version of the manuscript that addresses the points raised during the review process.

* A letter that responds to each point raised by the editor and reviewer(s). You should upload this letter as a separate file labeled 'Response to Reviewers'. This file does not need to include responses to any formatting updates and technical items listed in the 'Journal Requirements' section below.'. This file does not need to include responses to any formatting updates and technical items listed in the 'Journal Requirements' section below.'. This file does not need to include responses to any formatting updates and technical items listed in the 'Journal Requirements' section below.'. This file does not need to include responses to any formatting updates and technical items listed in the 'Journal Requirements' section below.

* A marked-up copy of your manuscript that highlights changes made to the original version. You should upload this as a separate file labeled 'Revised Manuscript with Track Changes'.'.'.'.

* An unmarked version of your revised paper without tracked changes. You should upload this as a separate file labeled 'Manuscript'.'.'.'.

We look forward to receiving your revised manuscript.

Kind regards,

Anand Setty Balakrishnan, PhD

Academic Editor

Qu Cheng

Section Editor

Shaden Kamhawi

co-Editor-in-Chief

Paul Brindley

co-Editor-in-Chief

**Journal Requirements:**

1) Please upload all main figures as separate Figure files in .tif or .eps format. For more information about how to convert and format your figure files please see our guidelines:

2) We have noticed that you have uploaded Supporting Information files, but you have not included a list of legends. Please add a full list of legends for your Supporting Information files after the references list.

**Reviewers' comments:**

Reviewer's Responses to Questions

**Key Review Criteria Required for Acceptance?**

**Methods**

-Are the objectives of the study clearly articulated with a clear testable hypothesis stated?

-Is the study design appropriate to address the stated objectives?

-Is the population clearly described and appropriate for the hypothesis being tested?

-Is the sample size sufficient to ensure adequate power to address the hypothesis being tested?

-Were correct statistical analysis used to support conclusions?

-Are there concerns about ethical or regulatory requirements being met?

Reviewer #4: (No Response)

Reviewer #5: -Objective clearly articulated

-The study design is appropriate to stated objective

-The population clearly described

-Sufficient sample size

-Well used

-No where ethical declaration stated

**Results**

-Does the analysis presented match the analysis plan?

-Are the results clearly and completely presented?

-Are the figures (Tables, Images) of sufficient quality for clarity?

Reviewer #4: (No Response)

Reviewer #5: -The integration of machine learning techniques with WASH and demographic data is timely, relevant and well presented

-Well presented

-not all Figures areclear. Lables of Figure 3-9 are like pictures (faintness)

**Conclusions**

-Are the conclusions supported by the data presented?

-Are the limitations of analysis clearly described?

-Do the authors discuss how these data can be helpful to advance our understanding of the topic under study?

-Is public health relevance addressed?

Reviewer #4: (No Response)

Reviewer #5: -Yes, source of the data well presented

-Limitations clearly stated in the recommendation section

-Not very much described

=Yes, the public health is well presented

**Editorial and Data Presentation Modifications?**

Reviewer #4: (No Response)

Reviewer #5: N/A

**Summary and General Comments**

Reviewer #4: (No Response)

Reviewer #5: This manuscript addresses an important public health issue in sub-Saharan Africa by examining NTD co-endemicity using machine learning integrated with WASH and demographic data, with clear relevance to Kenya’s NTD Elimination Strategic Plan 2030; however, substantial revisions are needed to strengthen clarity, rigor, and reproducibility. While the study is well motivated and empirically rich, it is affected by widespread language and formatting errors, inconsistencies in terminology and section numbering, unclear figure and table descriptions, and minor deviations from journal style. More critically, the moderate predictive performance of the models (AUC ≈ 0.62–0.70), low specificity, and their implications for public health decision-making are insufficiently discussed. The epidemiological rationale for constructing the co-endemicity outcome variable is not well justified, and class frequencies are not reported. In addition, the use of mean imputation for missing data may introduce bias and requires stronger justification and sensitivity analysis. Clearer differentiation between tuned and final models, improved explanation of counterintuitive correlations, reduced repetition of acronyms, and explicit clarification that correlation does not imply causation are necessary to enhance the manuscript’s scientific credibility and policy relevance.

PLOS authors have the option to publish the peer review history of their article (what does this mean?). If published, this will include your full peer review and any attached files.). If published, this will include your full peer review and any attached files.). If published, this will include your full peer review and any attached files.). If published, this will include your full peer review and any attached files.

...

Reviewer #4: No

Reviewer #5: No

**Figure resubmission:**
---

## [Editor Report · Decision Letter 2]

18 Mar 2026

Dear Ms MULWA,

We are pleased to inform you that your manuscript 'Advanced Machine Learning Approaches for Predicting Neglected Tropical Disease Co-endemicity in Kenya: A Focus on Soil-Transmitted Helminths, Schistosomiasis, and Lymphatic Filariasis.' has been provisionally accepted for publication in PLOS Neglected Tropical Diseases.

Best regards,

Anand Setty Balakrishnan, PhD

Academic Editor

Qu Cheng

Section Editor

Shaden Kamhawi

co-Editor-in-Chief

Paul Brindley

co-Editor-in-Chief

---

## [Editor Report · Acceptance letter]

Dear Ms MULWA,

We are delighted to inform you that your manuscript, "Advanced Machine Learning Approaches for Predicting Neglected Tropical Disease Co-endemicity in Kenya: A Focus on Soil-Transmitted Helminths, Schistosomiasis, and Lymphatic Filariasis.," has been formally accepted for publication in PLOS Neglected Tropical Diseases.

Best regards,

Shaden Kamhawi

co-Editor-in-Chief

Paul Brindley

co-Editor-in-Chief
